# Elevated Extracellular HSP72 and Blunted Heat Shock Response in Severe COVID-19 Patients

**DOI:** 10.3390/biom12101374

**Published:** 2022-09-26

**Authors:** Mariana Kras Borges Russo, Lucas Stahlhöfer Kowalewski, Gabriella Richter da Natividade, Carlos Henrique de Lemos Muller, Helena Trevisan Schroeder, Patrícia Martins Bock, Layane Ramos Ayres, Bernardo Urbano Cardoso, Caroline Zanotto, Julia Tsao Schein, Tatiana Helena Rech, Daisy Crispim, Luis Henrique Canani, Rogério Friedman, Cristiane Bauermann Leitão, Fernando Gerchman, Mauricio Krause

**Affiliations:** 1Laboratory of Inflammation, Metabolism and Exercise Research (LAPIMEX) and Laboratory of Cellular Physiology, Department of Physiology, Institute of Basic Health Sciences, Universidade Federal do Rio Grande do Sul, Porto Alegre 91509-900, RS, Brazil; 2Endocrine and Metabolic Unit, Hospital de Clinicas de Porto Alegre, Post-Graduate Program in Medical Sciences: Endocrinology, Universidade Federal do Rio Grande do Sul, Porto Alegre 91509-900, RS, Brazil; 3Post-Graduate Program in Medical Sciences: Endocrinology, Universidade Federal do Rio Grande do Sul, Porto Alegre 91509-900, RS, Brazil; 4Faculdades Integradas de Taquara, Taquara 95612-150, RS, Brazil; 5Department of Pharmacology, Universidade Federal do Rio Grande do Sul, Porto Alegre 91509-900, RS, Brazil; 6Intensive Care Unit, Hospital de Clinicas de Porto Alegre, Porto Alegre 90035-903, RS, Brazil

**Keywords:** SARS-CoV-2, inflammation, heat shock response, HSP72, metabolic diseases, critically ill patients

## Abstract

Aims: We hypothesized that critically ill patients with SARS-CoV-2 infection and insulin resistance would present a reduced Heat Shock Response (HSR), which is a pathway involved in proteostasis and anti-inflammation, subsequently leading to worse outcomes and higher inflammation. In this work we aimed: (i) to measure the concentration of extracellular HSP72 (eHSP72) in patients with severe COVID-19 and in comparison with noninfected patients; (ii) to compare the HSR between critically ill patients with COVID-19 (with and without diabetes); and (iii) to compare the HSR in these patients with noninfected individuals. Methods: Sixty critically ill adults with acute respiratory failure with SARS-CoV-2, with or without diabetes, were selected. Noninfected subjects were included for comparison (healthy, *n* = 19 and patients with diabetes, *n* = 22). Blood samples were collected to measure metabolism (glucose and HbA1c); oxidative stress (lypoperoxidation and carbonyls); cytokine profile (IL-10 and TNF); eHSP72; and the HSR (in vitro). Results: Patients with severe COVID-19 presented higher plasma eHSP72 compared with healthy individuals and noninfected patients with diabetes. Despite the high level of plasma cytokines, no differences were found between critically ill patients with COVID-19 with or without diabetes. Critically ill patients, when compared to noninfected, presented a blunted HSR. Oxidative stress markers followed the same pattern. No differences in the HSR (extracellular/intracellular level) were found between critically ill patients, with or without diabetes. Conclusions: We demonstrated that patients with severe COVID-19 have elevated plasma eHSP72 and that their HSR is blunted, regardless of the presence of diabetes. These results might explain the uncontrolled inflammation and also provide insights on the increased risk in developing type 2 diabetes after SARS-CoV-2 infection.

## 1. Introduction

COVID-19 (coronavirus disease 2019) ranges in severity from asymptomatic to acute respiratory distress syndrome (ARDS). The latter requires intensive care admission and mechanical ventilation and is consequently associated with a high mortality rate [1,2,3,4].

Critically ill patients with SARS-CoV-2 infection present a hypercoagulability state along with a syndrome called “cytokine storm” (as represented by the high pro-inflammatory state) [5]. Cytokine storm is a term applied to maladaptive cytokine release in response to viral infection, leading to elevated systemic levels of cytokines and chemokines, including: interleukin 1α (IL-1α); tumor necrosis factor alpha (TNF); interleukin 7 (IL-7); vascular endothelial growth factor (VEGF); interleukin 8 (IL-8); interferon gamma (IFN-γ); interleukin 9 (IL-9); interleukin 10 (IL-10); monocyte chemoattractant protein-1 (MCP-1); and others [6]. The elevated levels of these pro-inflammatory cytokines play a significant role in the morbidity and mortality of SARS-CoV-2 infections [6]. At the center of this storm is the nuclear factor kappa-light-chain-enhancer of activated B cells (NF-κB), which controls the expression of several proinflammatory proteins. In fact, in critically ill patients with COVID-19, the NF-κB signaling pathway is upregulated [7]. In general, the inhibition of the NF-κB signaling pathway may be mandatory for the proper resolution of inflammation in SARS-CoV-2-infected patients.

Resolution of inflammation can be reached through the activation of the heat shock transcription factor-1 (HSF1), thereby initiating a very conserved transcriptional program called the Heat Shock Response (HSR) [8]. Infection-induced elevation on the body temperature (fever) is a known inductor of the HSF-1 activation. The activation of the HSR induces the expression of Heat Shock Proteins (HSP), particularly the 70 kDa family (HSP70, and its inducible form HSP72), a protein with anti-inflammatory and cytoprotective proprieties [9,10].

The HSR (represented by HSP72 expression and release) is essential to protect the cells against a wide range of nonlethal stresses, such as oxidative stress, hyperthermia, exertional stress, exercise, ischemia, and metabolic stress [11]. HSP72 is mandatory to maintain cellular proteostasis by acting as a classical molecular chaperone [12]. In addition to its key role in the maintenance of proteostasis, HSP72 exerts a potent anti-inflammatory effect [13]. HSP72 can interact with the complex formed by NF-κB and its inhibitor (IκB), impeding NF-κB translocation to the nucleus, and thus decreasing its activity [14]. A detailed description of the HSR is available elsewhere [15]. Interestingly, the obesity-related chronic inflammatory state shows a depressed HSR, and the mechanisms for such findings are related to insulin resistance [16,17,18,19]. Thus, in SARS-CoV-2-infected people with some degree of insulin resistance, the lower HSR may partially explain the hyper-inflammatory state and the worse prognosis (when in comparison with insulin-sensitive infected subjects). This abnormal response might be even more pronounced in subjects with established type 2 diabetes.

The extracellular HSP72, opposite to its action in the intracellular environment, activates several proinflammatory responses. eHSP72 has been reported to stimulate neutrophil microbicidal capacity [20] and chemotaxis [21], recruitment of NK (natural killer) cells [22], as well as cytokine release from various immune cells [23,24]. In the extracellular compartment, this protein binds to cell-surface receptors known as the Toll-like receptor (TLR 2 until 4) [25]. This interaction can lead to the activation of proinflammatory signaling proteins such as MyD88 and TIRAP (which activate IKK, p38, JNK, and ultimately NF-κB), and induce changes in gene expression [26]. 

We have hypothesized (when determining worse outcomes) that critically ill patients with both SARS-CoV2 infection and insulin resistance would present a blunted HSR and, consequently, a higher level of inflammation [15]. In this work, we aimed: (i) to measure the basal concentration of extracellular HSP72 (eHSP72) in critically ill patients with severe COVID-19 pneumonia to compare with noninfected individuals, (ii) to compare the HSR (in vitro) between critically ill severe COVID-19 pneumonia patients with and without type 2 diabetes, and (iii) to compare HSR in these patients with noninfected individuals. 

## 2. Methods and Materials

### 2.1. Study Design and Participants

This study protocol was approved by the Ethics Committee of Hospital de Clínicas de Porto Alegre (CAAE 32962620600005327 and FIPE-HCPA 2020-0218). The study procedures were conducted according to the Declaration of Helsinki, and the informed consent was obtained from the patient’s legal representatives.

This is a prospective cohort study. Patients with acute respiratory failure admitted to the intensive care unit (ICU) of Hospital de Clinicas de Porto Alegre (Porto Alegre, Brazil) were prospectively included after screening from November 2020 to August 2021. Patients were eligible if they had laboratory-confirmed SARS-CoV-2 infection that was determined via reverse transcriptase–polymerase-chain reaction (RT-PCR) assay from either nasal or pharyngeal swabs. Inclusion criteria were determined via ICU admission that was less than 48 h, orotracheal intubation and mechanical ventilation (MV) within the first 48 h of ICU admission, and age ≥18 years. The exclusion criteria were chronic kidney disease on dialysis; cirrhosis Child–Pugh B or C; chronic corticosteroid use; hypercortisolism; adrenal insufficiency; solid organ transplantation; gastric surgery or small bowel resection (including bariatric surgery); decreased intestinal absorption; life expectancy less than 24 h; pregnancy or breastfeeding; and participation of interventions groups from other studies. 

In order to compare the obtained results from critically ill patients with severe COVID-19 pneumonia with noninfected subjects, we used the data and stored frozen samples from previous studies [17,27,28]. These samples were obtained from healthy individuals (here named noninfected control group, *n* = 19) and individuals with type 2 diabetes (noninfected with diabetes, *n* = 22). All data analyses were performed using the same methodology. 

### 2.2. Procedures and Biochemistry Measurements

All consecutive patients over 18 years of age admitted to the ICU with a confirmed SARS-CoV-2 RT-PCR test and submitted to MV within the first 48 h of ICU admission were eligible for study entry. 

Blood samples were collected until 72 h of ICU admission. The baseline characteristics such as age, gender, comorbidities, and medications were collected from electronic medical records and from the patient’s family when needed. Diabetes was defined based on previous diagnosis, current use of anti-hyperglycemic medications, or an admission glycated hemoglobin (A1C) value ≥6.5% (48 mmol/mol).

Blood samples were obtained from central catheters and stored in heparin-coated and gel-clotted Vacutainer^TM^ tubes using standard aseptic techniques. Samples were immediately centrifuged (at 4 °C and 1000× *g* for 15 min), after which plasma and serum were removed and stored at −80 °C for further analysis. Plasma glycemia levels were measured in an automated system Cobas C111 (Roche Diagnostics, Basel, Switzerland). HbA1c was measured by HPLC (Variant II Turbo) and expressed as a % of total hemoglobin for HbA1c.

### 2.3. Plasma Cytokine Quantification

Blood samples were collected in EDTA tubes and centrifuged immediately at 4 °C and 1000× *g* for 15 min. The separated plasma was stored at −80 °C until analysis. Plasma values of TNF and IL-10 were assessed by magnetic bead assay using the Human Magnetic Custom Luminex^®^ Kit (Invitrogen Life Technologies, Carlsbad, CA, USA) and the Luminex^®^ 200TM magnetic bead plate reader (Luminex, Austin, TX, USA) following the manufacturers’ instructions.

A standard curve was generated by serial dilutions of the reconstituted standard. Samples and standards were incubated with mixed beads overnight at room temperature on an orbital shaker. Beads were washed and then incubated with the detection antibodies at room temperature for 1 h and with streptavidin for 30 min. Beads were washed and resuspended, and the plate was subsequently analyzed on the Luminex^®^ 200TM reader. The results were plotted as a function of fluorescence intensity. Mean fluorescence intensity (MFI) takes into account the number of fluorescent pixels within the scanned area. MFI values below the detection limit were assumed to be missing values. MFI was then converted to picograms (pg)/mL based on the standard curve. All samples were analyzed in duplicate.

### 2.4. Oxidative Damage

Thiobarbituric acid-reactive substances (TBARS) were used to determine lipid peroxidation. Briefly, samples were first centrifuged at 12,000× *g* at 4 °C for 10 min, then 250 μL of the sample, 10 μL of 4.5 mM butylated Hydroxytoluene (BHT), and 200 μL of 30% trichloroacetic acid (TCA) were added to 1.5 mL Eppendorf tubes. These were subsequently placed in a boiling water bath (100 °C) for 15 min, and centrifuged at 15,000× *g* at room temperature for 2 min. Next, 400 μL of supernatant and 400 μL of 0.23% thiobarbituric acid (TBA) were pipetted into the cryotubes and boiled in a 100 °C water bath for 30 min. The samples were cooled down for 5 min and pipetted in duplicates of 200 μL into a 96-well plate. TBARS was then determined in a microplate reader at 540 nm (Multiskan Go, Thermo Scientific, Waltham, MA, USA) [29]. Carbonyl assay was used to determine oxidative damage to proteins and the absorbance was read at 360 nm (Multiskan Go, Thermo Scientific, Waltham, MA, USA) [30]. 

### 2.5. Heat Shock Response Test

Considering the importance of the HSR for stress adaptation, we tested the capacity of peripheral blood mononuclear cells (PBMCs) (a major source of circulating HSP72 and representative of the immune cell stress response), to release HSP72, under heat stress conditions (a normal and expected response in healthy cells). Briefly (accordingly with the protocol [19]), after harvesting, the whole blood was immediately incubated at two different temperatures: 37 °C (control) and 42 °C (heat stressed) for 2 h in a water bath (with a gentle mix every 15 min). After incubation, the total blood was centrifuged to isolate plasma/serum and the PBMCs through a density gradient separation, as previously described [31]. Then, plasma was used for the direct analysis of extracellular HSP72, while PBMCs were prepared for the measurement of iHSP72. The PBMCs were washed and treated to ensure the absence of erythrocytes. PBMCs were then resuspended in an RPMI 1640 medium (pH 7,4 supplemented with 2% NaHCO3, 10% bovine calf serum, 100 U/mL penicillin and 100 µg/mL streptomycin); seeded in a 24-well flat-bottom plate (1 × 10^6^ cells/well); and placed in an incubator for 6 h (37 °C in a 5% CO2) in order to recover from the HS and reach the peak of HSP70 expression [19]. Cells were then removed from the incubator, lysed, and the total content of proteins was prepared for Western blotting analysis [19]. The difference between the concentration at 37 °C and 42 °C is used as an HSR index [19]. This test was applied previously, in several different conditions and diseases [17,32]. 

### 2.6. Extracellular HSP72 Quantification

A highly sensitive enzyme-linked immunosorbent assay (ELISA) method (EKS-715, Stressgen, Victoria, BC, Canada) was used to quantify the levels of plasma HSP72 protein as previously described [33]. Absorbance was measured at 450 nm and a standard curve was constructed from known dilutions of HSP72 protein to allow quantitative assessment of HSP72 concentration. Quantifications were made using a microplate reader (Multiskan Go, Thermo Scientific, Waltham, MA, USA). 

### 2.7. Protein Quantification and Western Blotting for Intracellular HSP70 Immunocontent

Cellular protein quantification was determined using a BCA Protein Assay Kit (Thermo Scientific, Waltham, MA, USA), and samples (1 µg) were mixed with 5× Laemmli loading buffer [50 mM Tris, 10% SDS, 10% glycerol, 10% 2-mercaptoethanol, and 2 mg/mL bromophenol blue] at a final concentration of 1:5, boiled for 5 min, and then electrophoresed [32]. For sodium dodecyl sulfate-polyacrylamide gel electrophoresis (SDS-PAGE), equivalent amounts of protein (1 µg) were applied in a 10% polyacrylamide minigel for 2 h at 100V [32]. Proteins were then transferred onto nitrocellulose membranes (GE Healthcare, Chicago, IL, USA) according to the manufacturer’s instructions (Bio-Rad, Hercules, FL, USA) (2 h, 100 V) [32]. For immunoblotting, membranes were blocked in 2% BSA in a wash buffer [50 mM Tris, 5 mM EDTA, 150 mM NaCl (TEN)-Tween 20 0.1% solution, pH 7.4] for 30 min and then incubated overnight at a 1:1.000 dilution with monoclonal Anti-HSP70 antibodies that were produced from mice (Sigma-Aldrich, St. Louis, MO, USA) [32]. After appropriate washing, the membranes were probed with anti-mouse IgG and biotin antibodies at a 1:10.000 dilution (Sigma-Aldrich, St. Louis, MO, USA) for 1h. Membranes were then incubated with a final Streptavidin−Peroxidase Polymer, Ultrasensitive (Sigma-Aldrich, St. Louis, MO, USA), at a 1:1.000 dilution for 1 h. Visualization of the blots was performed using the chemiluminescence reagent, p-coumaric acid, and luminol in an ImageQuantTM LAS 4000 chemiluminescence system (GE Healthcare, Chicago, IL, USA), and quantified using ImageJ (version 1.51f; NIH, Maryland City, MD, USA) [32]. A standard molecular weight marker (RPN 800, Rainbow Full Range Bio-Rad, CA, USA) was used as a reference to determine the molecular weights of the bands. The data were then normalized using GAPDH expression. 

### 2.8. Statistical Analysis and Sample Size

Categorical variables were expressed as percentages. Data were expressed as mean ± standard deviation (SD) or median [P25-P75], depending on variable distribution. Groups were compared using one-way analysis of variance with the Bonferroni post hoc test, the Kruskal–Wallis test, or chi-square test, as appropriate. Values were considered statistically significant if *p* < 0.05. Statistical analyses were performed using SPSS, version 23.0 (Armonk, NY, USA). Further, Spearman’s rank order correlation coefficient (r) was used to determine correlations between extracellular HSP72 HSR and other variables. For all analyses, statistical significance was accepted for *p* < 0.05. Our samples were collected from a smaller group of patients selected from a major study. The sample size was calculated to be 420 patients using an EnvStats package, version 2.3.1, R software, to detect a difference in mortality between groups (controls vs. people with diabetes)—with a power of 80% and an α-error of 5%—and considering 10% to be missing due to dosage errors [34]. From this population, we collected blood samples from 60 consecutive patients. 

## 3. Results

### 3.1. Patient Characteristics and Cytokine Profile in Critically Ill Patients with Severe COVID-19 Pneumonia

The baseline characteristics of critically ill patients with severe COVID-19 pneumonia are described in Table 1. No significant differences were found for age, weight, and body mass index (BMI) between the critically ill patients (with vs. without type 2 diabetes). As expected, glycaemia and HbA1C were higher in patients with diabetes. No statistical differences were observed between the groups for TNF (26.6 [9.36–32.39] vs. 17.7 [12.77–27.41] pg/mL), IL-10 (2.86 [1.42–4.57] vs. 2.0 [1.42–6.87] pg/mL), or TNF/IL-10 (6.85 [3.35–13.28] vs. 5.06 [2.38–10.43]).

### 3.2. Plasma Extracellular HSP72 Concentration in Critically Ill Patients with COVID-19 

As depicted in Figure 1, no differences between critically ill patients with COVID-19 were found for eHSP72 (0.453 ± 0.202 vs. 0.547 ± 0.253 ng/mL). Interestingly, we detected a positive correlation between plasma eHSP72 and HbA1C (*r* = 0.394, *p* = 0.0042) in critically ill patients with COVID-19.

### 3.3. Comparison of Plasma HSP72 among Different Groups: Noninfected Control Subjects, Noninfected Subjects with Diabetes and Critically Ill Patients with COVID-19 Pneumonia

Since we were expecting much higher levels of eHSP72 in critically ill patients with COVID-19, and as there are no available data in the literature, we decided to compare the results of eHSP72 in critically ill patients with other groups of patients from previous studies, including: noninfected healthy controls (*n* = 19) and noninfected subjects with diabetes (*n* = 22). For this comparison among the three groups, we analyzed all critically ill patients together, both those with and without diabetes (Figure 2). 

Critically ill patients with COVID-19 pneumonia presented higher plasma concentrations of eHSP72 (0.486 ± 0.23 ng/mL) when compared with noninfected healthy controls (0.096 ± 0.06 ng/mL) and noninfected subjects with diabetes (0.198 ± 0.05 ng/mL). 

### 3.4. Heat Shock Response in Critically Ill Patients with Severe COVID-19 Pneumonia 

As shown in Figure 3, no differences in the HSR were found between critically ill patients with COVID-19 with or without diabetes. To confirm that both the exocytosis and intracellular content of HSP72 were reduced, we analyzed the immunocontent of HSP72 from the PBMCs. As depicted in Figure 3, intracellular quantification confirmed the reduced expression of iHSP72. A possible inverse correlation between intracellular HSR and the inflammatory index (TNF-α/IL-10) (*r* = −0.284) may be present, despite not being statistically significant (*p* = 0.058). 

### 3.5. Comparison of Heat Shock Response between Different Groups: Noninfected Control Subjects, Noninfected Subjects with Controlled Diabetes, and Critically Ill Patients with Severe COVID-19 Pneumonia

To compare the HSR with other groups, we analyzed blood from noninfected patients: the healthy control group and the noninfected subjects with diabetes. In a different approach from the baseline blood eHSP72 group, we included all groups that we performed the HSR test on: noninfected controls, noninfected subjects with diabetes, critically ill patients with COVID-19 without diabetes, and critically ill patients with COVID-19 with diabetes. Table 2 shows the patient group characteristics. Notably, infected patients even without a previous diagnosis of diabetes and HbA1c <6.5% presented significant hyperglycemia, most likely due to stress hyperglycemia (>140 mg/mL). In line with this, infected patients with diabetes also had higher serum glucose values when compared to noninfected subjects with diabetes. Interestingly, obesity was highly prevalent in critically ill patients with COVID-19, regardless of the presence of diabetes, as demonstrated by the higher BMI observed than usually seen in the critically ill population (Table 2). This may indicate the presence of insulin resistance and may also explain the hyperglycemia and the levels of HbA1c within this group.

Figure 4 depicts the comparison between noninfected healthy controls, noninfected subjects with diabetes, and critically ill patients with COVID-19 pneumonia (with or without diabetes). As previously compared, patients without diabetes have a preserved HSR [18]. However, in the presence of COVID-19 infection, particularly in critically ill patients, the HSR is blunted even in the absence of diabetes. Thus, a similarly disrupted HSR can be found in noninfected subjects with diabetes and in critically ill patients with COVID-19. Regarding markers of oxidative damage (lipids and proteins) our results have shown that, compared to healthy people, infected subjects present higher levels of damage (Figure 5). However, following the same pattern of the HSR, no differences exist between infected controls and diabetics.

## 4. Discussion

This is the first study that measured the basal concentration of blood HSP72 and the HSR status in critically ill patients with severe COVID-19 pneumonia, with and without diabetes. The main findings of this study are: (1) the concentration of plasma eHSP72 is high in critically ill patients with COVID-19 pneumonia and (2) the HSR is blunted in critically ill patients with severe COVID-19 pneumonia when compared to healthy patients without COVID-19, regardless of their glycemic status.

In conditions where chronic inflammation and oxidative stress occur, such as in patients with type 1 diabetes (T1DM) [35] and type 2 diabetes (T2DM), higher levels of eHSP72 are also present [36]. In fact, serum eHSP72 concentration is positively correlated with markers of inflammation in humans, such as C-reactive protein, monocyte count, and TNF-α [37,38]. However, in this sample of critically ill patients, the presence of diabetes did not influence the eHSP72 levels. These results indicate that, apparently, metabolic disease does not seem to influence the HSR when the patient is severely ill. As the blunted HRS is similar in critically ill patients with COVID-19 (with or without diabetes) and noninfected patients with diabetes, we may state that diabetes induces similar responses in terms of HSP72 and the HSR systems that severe COVID-19 (by different mechanisms) induces. In the same vein, in the past, diabetes was believed to be a cardiovascular equivalent [39]. Since the HSR is already significantly depressed in subjects with diabetes, no further decrease is observed in these patients when facing severe COVID-19. 

Another complication related to the elevated levels of eHSP72 is insulin resistance [33]. The underlying mechanisms that may lead to insulin resistance could involve an eHSP70-mediated stimulation of the TLR2/4. Accordingly, the TLR2/4-dependent activation of JNKs promotes phosphorylation of IRS-1 at Ser307 in rodents (equivalent to Ser312 in humans), leading to the inhibition of Akt activation [40], to a reduced glucose uptake by sensitive tissues, and to a state of resistance to insulin action. Thus, higher blood eHSP72 levels in patients with COVID-19 might contribute to the insulin resistance and stress hyperglycemia commonly present in these patients (Table 2), which may lead to negative outcomes. In addition, there is also a possible deleterious effect of elevated eHSP72 on pancreatic beta cells. Previous studies have shown that chronic high eHSP72 exerts direct effects on clonal pancreatic human and rodent beta cells and islets, such as decreased beta cell and islet viability, insulin secretion, and mitochondrial function [33]. Taken together, the elevated eHSP72 concentration may be related to the long-term metabolic imbalance reported in the literature [41,42,43]. 

Regarding the HSR, our initial hypothesis was that critically ill patients with COVID-19 pneumonia with diabetes would have a lower response when compared to those critically ill patients without diabetes, leading to better outcomes in the last group. However, we found that, in critically ill patients with COVID-19, the HSR is blunted regardless of the presence of diabetes, indicating that the virus may directly inhibit this pathway or, perhaps, use all chaperone machinery for its own cycle of replication (Figure 6).

In different cell types, the higher demand for protein synthesis may lead to the unfolded protein response (UPR) and endoplasmic reticulum stress (ER stress) [13]. The UPR is triggered to avoid the formation of protein aggregates that cause cellular dysfunction. If the UPR is resolved through autophagy and the HS response, then cells can reach proteostasis again, otherwise, they undergo apoptosis [14]. HSP70 is mandatory to maintain proteostasis during high demand of protein synthesis, as it occurs in beta cells producing high insulin content [13]. Under normal conditions, when HSP72 expression is normal, the chaperone supply for the appropriate folding of newly synthetized proteins is sufficient. However, due to an inflammation-induced inability to express iHSP70, the UPR is induced, causing cell dysfunction, perpetuation of inflammation, and apoptosis [13]. Viral activities have a profound impact on ER function [46]. In particular, SARS-CoV hijacks the ER to process its structural and nonstructural proteins [47,48]. In fact, it was demonstrated, in vitro, that the induction of ER stress and the UPR by SARS-CoV is modulated through the S protein [47]. In that paper, using kidney epithelial cells, the authors have shown that the S protein modulated ER stress differentially by stimulating PKR-like ER kinase (PERK), but sparing the other two branches of the UPR signaling mediated through IRE-1 and ATF-6 (inositol-requiring enzyme 1 and activating transcription factor 6), respectively [47]. Thus, it is reasonable to speculate that SARS-CoV-2 overloads the ER to produce the necessary material for replication, leading to an increased use of chaperones (HSP72, for example), and finally causing the full depletion of this pathway. Our findings suggest HSP72 to be lower than expected, independent from the previous metabolic state. Without HSP72, cells would lose their proteostasis control and leave inflammatory factors, such as NF-κB, free to increase the inflammation in a positive forward feedback system. In fact, the lack of a proper HSR in critically ill patients with COVID-19 might explain, at least in part, the high levels of proinflammatory cytokines in our cohort, which was found to be without a difference between patients with or without diabetes. Despite not being statistically significant (*p* = 0.058), we found a possible inverse correlation between an intracellular HSR and the inflammatory index (TNF-α/IL-10) (*r* = −0.284).

The HSR is essential for the maintenance of proteostasis and the inflammatory status. Stress-activated HSF1 can provide a robust anti-inflammatory response through the induction of HSP72. However, a mandatory pathway to maintain a normal chaperone machinery (i.e., in the HSR) is through insulin signaling [11]. Hampered insulin signaling will lead to a deficient ability to induce the HSR in order to resolve inflammation (which is a mechanism associated with overactivity of the GSK-3β enzyme). Not surprisingly, obesity-related chronic inflammatory states show a depressed HSR [16,17,18]. Thus, a lower HSR in insulin-resistant individuals might be responsible for the exacerbated levels of inflammation and the comparatively worse prognosis observed in critically ill patients infected by SARS-CoV-2, perhaps as little is known regarding insulin signaling during critical illness [49]. In addition, infection caused by SARS-CoV-2 may lead to downregulation of ACE2. This would decrease the production of angiotensin-(1-7) and the activation of the MAS receptors. MAS receptor activation can induce anti-inflammatory responses and, through the activation of SIRT1, stimulate the HSR pathway [44]. Without this axis, the HSR is diminished and the inhibitory effect of intracellular HSP72 over NF-κB is absent, leading to a disrupted inflammatory resolution (Figure 6).

Considering the therapeutic potential that increasing the HSR may have on the inflammatory response in critically ill patients with severe COVID-19 pneumonia, we could suggest different strategies of treatment: (i) the rational use of antipyretic drugs (allowing mild elevation of body temperature by fever, without causing heat damage, but guaranteeing the increase in HSP72); (ii) artificially increasing the body temperature (using thermal blankets, for example (please refer to reference [45] to understand the beneficial effects of heat therapy); and (iii) the use of HSP72 activators such as BGP-15 [15]. The pharmacological inducer of HSP72 and BGP-15, has been shown to be safe and well tolerated in phase II clinical trials in patients with diabetes and insulin resistance [50,51]. In addition, the use of BGP-15 in animal models was found to induce metabolic benefits, as well as reducing inflammatory signaling, and improving respiratory muscles during mechanical ventilation [52]. Nevertheless, the efficiency and safety of such therapies could be tested in this study population.

This study has limitations. First, despite the fact that our data show no difference in the HSR between critically ill patients with SARS-CoV-2 infection with or without diabetes, it is important to consider that most of our critically ill patients had obesity, with potentially high degrees of insulin resistance. For this reason, we need to look at the data with caution, since our study population without diabetes were not “metabolic healthy”. Second, the lack of information regarding the HSR in noncritically ill patients with COVID-19 prevents us from testing if the previous level of the HSR would determine the evolution of the disease and the need for ICU admission. Finally, low sample size, inability to stratify data by sex, and the inability to account for previous medications were also a limitation in this work. Additional information regarding patients medications and full data can be found at Appendix A.

## 5. Conclusions

Despite the limitations of this study, we demonstrated that critically ill patients with severe COVID-19 pneumonia present elevated concentrations of plasma eHSP72. In addition, the HSR, a vital pathway for proteostasis and anti-inflammation, is blunted. The consequences of these two abnormalities might explain, in part, the uncontrolled inflammation (cytokine storm) seen in these patients. Finally, the administration of HSR activators should be further investigated. 

## Figures and Tables

**Figure 1 biomolecules-12-01374-f001:**
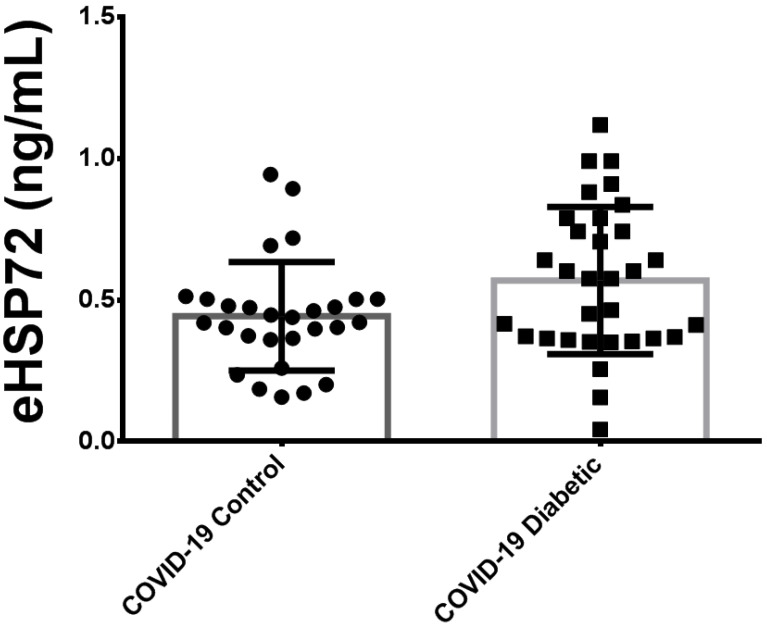
Comparison of plasma concentration of eHSP72 between critically ill patients with COVID-19 pneumonia with and without diabetes.

**Figure 2 biomolecules-12-01374-f002:**
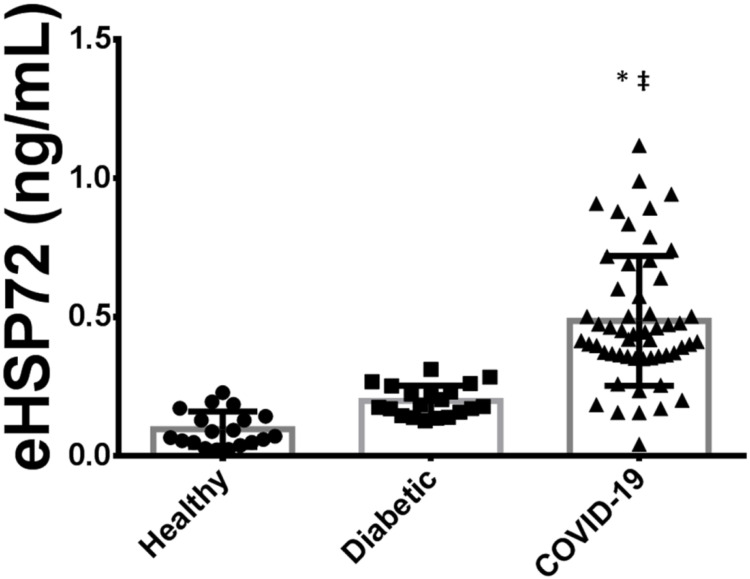
Comparison of plasma concentration of eHSP72 between critically ill patients with COVID-19 pneumonia, noninfected healthy subjects and noninfected patients with diabetes. * When different from noninfected controls (healthy). ‡ When different from noninfected patients with diabetes *p* < 0.05. Data expressed as mean ± SD.

**Figure 3 biomolecules-12-01374-f003:**
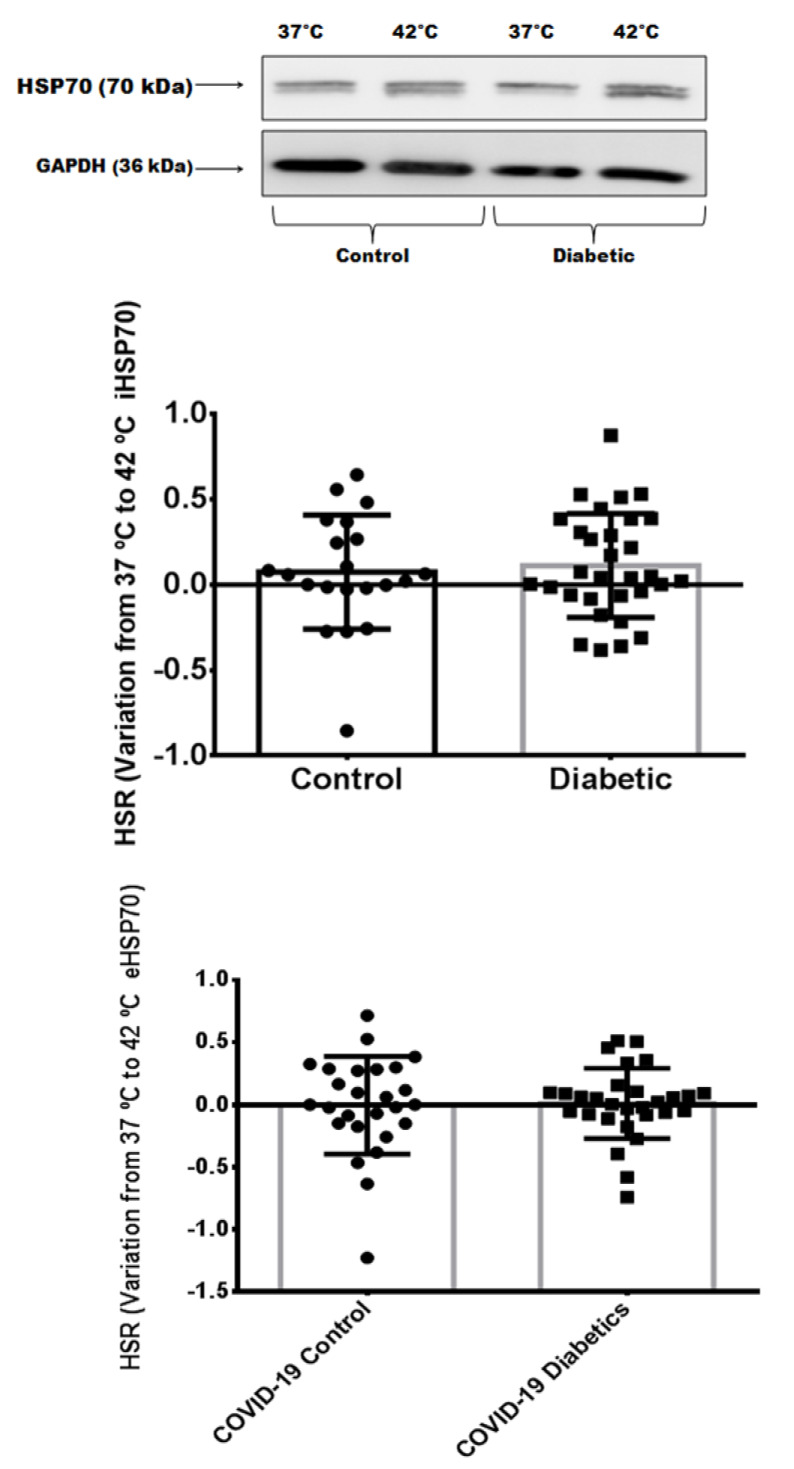
Comparison of Heat Shock Response (intracellular) between critically ill patients with COVID-19 pneumonia with and without diabetes. Briefly, the collected blood was immediately incubated (whole blood) at two different temperatures: 37 °C (control) and 42 °C (heat stressed) for 2 h in a water bath. After the incubation, the PBMC were resuspended in an RPMI 1640 medium (pH 7.4 supplemented with 2% NaHCO_3_, 10% bovine calf serum, 100 U/mL penicillin and 100 µg/mL streptomycin), seeded in a 24-well flat-bottom plate (1 × 10^6^ cells/well), and placed in an incubator for 6 h (37 °C in a 5% CO_2_) in order to recover from the HS and reach the peak of HSP70 expression. Cells were then removed from the incubator, appropriately lysed and the total content of proteins was then prepared for Western blotting analysis. Plasma was collected and used for HSP72 measurements. The difference between concentration at 37 °C and 42 °C is used as the HSR index. A representative image was used to show the immunocontent of iHSP72. Data expressed as mean ± SD.

**Figure 4 biomolecules-12-01374-f004:**
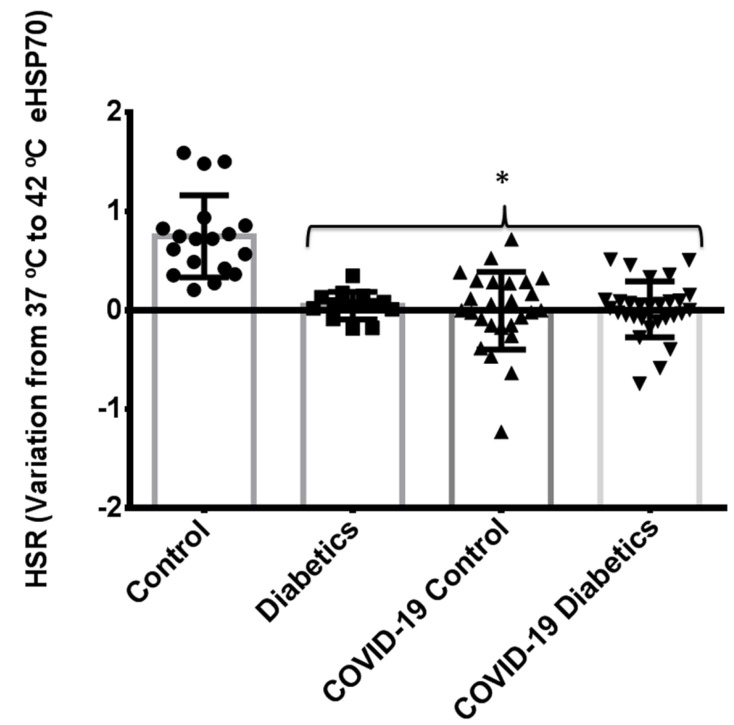
Comparison of the Heat Shock Response between critically ill patients with COVID-19 pneumonia with and without diabetes, noninfected healthy subjects, and noninfected patients with diabetes. Briefly, the collected blood, was immediately incubated (i.e., the whole blood) at two different temperatures: 37 °C (control) and 42 °C (heat stressed) for 2 h in a water bath. After the incubation, plasma was collected and used for HSP72 measurements. The difference between concentration at 37 °C and 42 °C is used as the HSR index. * When different from noninfected healthy subjects. *p* < 0.05. Data expressed as mean ± SD.

**Figure 5 biomolecules-12-01374-f005:**
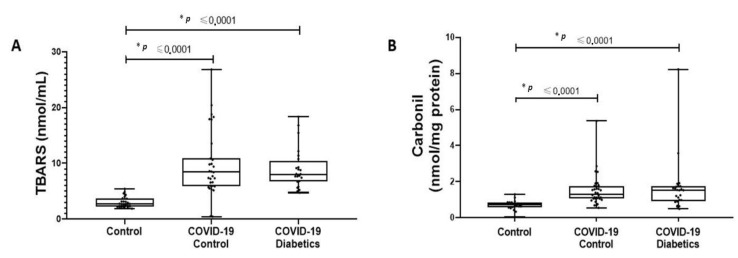
Comparison of lipid (**A**) and protein (**B**) oxidative damage between critically ill patients with COVID-19 pneumonia with and without diabetes, noninfected healthy subjects. * When different from noninfected healthy subjects. *p* < 0.05. Data expressed as mean ± SD.

**Figure 6 biomolecules-12-01374-f006:**
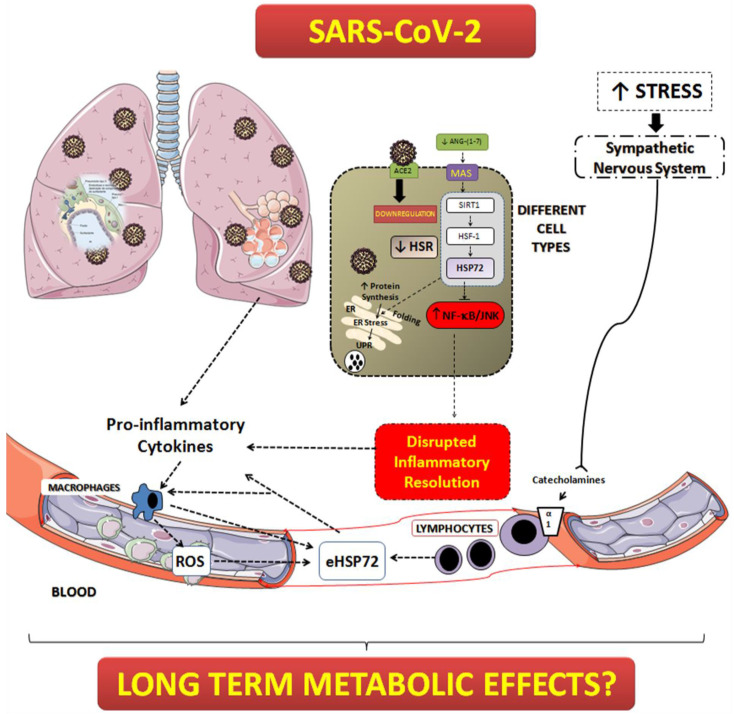
Possible role of elevated plasma eHSP70 and a lower HSR in a SARS-CoV-2 infection. In several cells, infection caused by SARS-CoV-2 might lead to downregulation of ACE2. This would decrease the production of angiotensin (1–7) and the activation of the MAS receptor. MAS receptor activation induces anti-inflammatory responses and, in addition, by the activation of SIRT1, stimulates the HSR pathway [44]. Without this axis, the HSR is diminished and the inhibitory effect of intracellular HSP72 over NF-κB is absent, leading to a disrupted inflammatory resolution. The cytokine storm induces direct damage on cells/tissues and activates, along with the increased sympathetic tonus, the release of HSP72 to the extracellular environment (eHSP72) [45]. Extracellular HSP72 might reinforce the inflammatory system and induce activation of TLR (Toll-like receptors, particularly TLR4) in several cell types, including pancreatic beta cells and skeletal muscle [11]. The chronic activation of the TLRs by eHSP72 might reduce the capacity of the HSR activation, worsening the inflammatory response and altering metabolic status.

**Table 1 biomolecules-12-01374-t001:** Subjects’ characteristics, biochemistry, and cytokine profile.

Patient Characteristic	Critically Ill Infected COVID-19: Control (*n* = 27)	Critically Ill Infected COVID-19: Diabetics (*n* = 32)	*p* Value
Sex (M/F)	(13/14)	(18/14)	
Age (years)	58.2 ± 13.5	63.5 ± 11.2	*p* = 0.522
Body Mass (kg)	90.8 ± 18.42	89.3 ± 20	*p* = 0.762
BMI (kg/m^2^)	33 ± 6.7	33.6 ± 7.9	*p* = 0.631
Glycemia (mg/dL)	171 ± 48	235 ± 79 ^#^	*p* = 0.006
HbA1C (%)	5.9 ± 0.5	8.9 ± 2.2 ^#^	*p* < 0.0001
TNF-α (pg/mL)	26.6 [9.36–32.39]	17.7 [12.77–27.41]	*p* = 0.532
IL-10 (pg/mL)	2.86 [1.42–4.57]	2.0 [1.42–6.87]	*p* = 0.682
TNF-α/IL-10	6.85 [3.35–13.28]	5.06 [2.38–10.43]	*p* = 0.405

^#^ When different from Critically Ill Infected SARS-CoV2: Control. Data expressed as absolute number, mean ± SD or median [P25-75].

**Table 2 biomolecules-12-01374-t002:** Subjects’ characteristics and general biochemistry: All groups.

Patient Characteristic	Noninfected Control (*n* = 19)	Noninfected Diabetics (*n* = 22)	Critically Ill Infected COVID-19: Control (*n* = 27)	Critically Ill Infected COVID-19: Diabetics (*n* = 32)
Age (years)	54.5 ± 8.3	68.9 ± 7.8	58.2 ± 13.5	63.5 ± 11.2
Body Mass (kg)	68.1 ± 9.3	79.12 ± 10.8	90.8 ± 18.42 *	89.3 ± 20 *
Height (m)	1.63 ± 0.08	1.66 ± 0.8	1.66 ± 0.1	1.67 ± 0.1
BMI (kg/m^2^)	25.6 ± 2.5	28.7 ± 3.1	33 ± 6.7 *	33.6 ± 7.9 *
Glycaemia (mg/dL)	102.5 ± 12.4	133.1 ± 21.4	171.1 ± 48.38 *^‡^	235.7 ± 79.4 *^‡^^#^
HbA1C (%)	-	6.75 ± 0.6	5.94 ± 0.51	8.9 ± 2.2 ^‡^^#^

* Different from noninfected controls. ^‡^ When different from noninfected diabetics. ^#^ When different from critically ill, infected COVID-19: Control. Data expressed as absolute number or mean ± SD.

## Data Availability

The data used to support the findings of this study are available from the corresponding authors upon request.

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
