# Peer review of "Elevated Extracellular HSP72 and Blunted Heat Shock Response in Severe COVID-19 Patients"

_biomolecules, 2022, doi:10.3390/biom12101374_

Round 1

Reviewer 1 Report

As the authors outlined, since HSR is already significantly depressed in subjects with diabetes, it is not really surprising, that no further decrease is observed in those of patients having severe COVID-19. Also it is obvious, that higher blood eHSP72 levels in patients with COVID-19 might contribute to the IR and stress hyperglycemia. Most likely, the eHSP70-mediated stimulation of TLR2/4 is the key and intitiating step in the underlying mechanism leading to IR, as was described in the menuscript. Taken together, indeed "diabetes is equivalent to severe COVID-19 in terms of HSP72 and HSR systems". 

The authors point to the clear therapeutic potential of known HSR modulation strategies to elevate HSR in critically ill patients with severe COVID-19 pneumonia. Namely, using antipyretic drugs (fever), increasing body temperature (heat therapies with different ways) or, activate HSR by HSP inducer drugs, drug candidates (like BGP-15). 

Minor comment: 

It was accepted for many years, that heat shock genes are basically upregulated by stress conditions by the formation of misfolded/denatured proteins. Thus, during nonstress conditions heat shock proteins (HSPs) are expressed at low levels and maintain the monomer heat shock factors (mostly  HSF 1)  in an inactive, repressed state. De-repression of HSFs occurs as a result of the titration of HSPs away from the HSFs by the stress-induced formation of denatured proteins. HSF then translocates into the nucleus, trimerizes, undergoes hyperphosphorylation and binds to hsp gene promoters, thus leading to the subsequent expression of their proteins. However, this “classics” view as a dogma did not fit many specific situations and alternative, though not exclusive models of stress sensing/signaling  have been suggested, as well.

In line of that, for instance a pioneering work has emerged from Murakami-Murofushi and co-workers laboratory: they firstly suggested, that cholesteryl glucoside (CG) plays a determining role in the control of heat shock response in various systems. This group furnished evidences  that indeed, CG is rapidly induced upon heat shock well before the occurrence of HSF1 activation and heat shock protein production. Moreover, exogeneous CG was shown to induce HSF1 activation and HSP70 upregulation in human fibroblasts. Taken together, they proposed that CG – together with several other candidate lipids – may function as a key lipid mediator in the stress protein responses of mammalian cells (See also in the literature the "Membrane lipid therapy").

Final comment: not only synthetic drugs, but also natural compounds can correct HSR via HSP activation (see for instance the principle of xenohormesis, i.e. those of bioactive compounds that can confer stress resistance)

Author Response

Comments and Suggestions for Authors

As the authors outlined, since HSR is already significantly depressed in subjects with diabetes, it is not really surprising, that no further decrease is observed in those of patients having severe COVID-19. Also it is obvious, that higher blood eHSP72 levels in patients with COVID-19 might contribute to the IR and stress hyperglycemia. Most likely, the eHSP70-mediated stimulation of TLR2/4 is the key and intitiating step in the underlying mechanism leading to IR, as was described in the menuscript. Taken together, indeed "diabetes is equivalent to severe COVID-19 in terms of HSP72 and HSR systems".

The authors point to the clear therapeutic potential of known HSR modulation strategies to elevate HSR in critically ill patients with severe COVID-19 pneumonia. Namely, using antipyretic drugs (fever), increasing body temperature (heat therapies with different ways) or, activate HSR by HSP inducer drugs, drug candidates (like BGP-15).

Minor comment:

It was accepted for many years, that heat shock genes are basically upregulated by stress conditions by the formation of misfolded/denatured proteins. Thus, during nonstress conditions heat shock proteins (HSPs) are expressed at low levels and maintain the monomer heat shock factors (mostly  HSF 1)  in an inactive, repressed state. De-repression of HSFs occurs as a result of the titration of HSPs away from the HSFs by the stress-induced formation of denatured proteins. HSF then translocates into the nucleus, trimerizes, undergoes hyperphosphorylation and binds to hsp gene promoters, thus leading to the subsequent expression of their proteins. However, this “classics” view as a dogma did not fit many specific situations and alternative, though not exclusive models of stress sensing/signaling  have been suggested, as well.

In line of that, for instance a pioneering work has emerged from Murakami-Murofushi and co-workers laboratory: they firstly suggested, that cholesteryl glucoside (CG) plays a determining role in the control of heat shock response in various systems. This group furnished evidences  that indeed, CG is rapidly induced upon heat shock well before the occurrence of HSF1 activation and heat shock protein production. Moreover, exogeneous CG was shown to induce HSF1 activation and HSP70 upregulation in human fibroblasts. Taken together, they proposed that CG – together with several other candidate lipids – may function as a key lipid mediator in the stress protein responses of mammalian cells (See also in the literature the "Membrane lipid therapy").

Final comment: not only synthetic drugs, but also natural compounds can correct HSR via HSP activation (see for instance the principle of xenohormesis, i.e. those of bioactive compounds that can confer stress resistance)

Author Response: Author Response: We thank the reviewer for the positive observation in regards to our study and the generated results. We also appreciate the excellent suggestions in terms of new mechanisms of HSR activation, particularly CG and the membrane lipid therapy. We are preparing a review article looking at the effects of alternative treatments to improve HSR (including exercise, heat therapy, etc) and we will include the reviewer suggestions.   

Reviewer 2 Report

The manuscript by Mariana Kras Borges Russo et al. aims to compare HSP levels and HSR in patients w/ and w/o severe Covid-19 syndrome; and to understand whether diabetes has an effect on HSP levels or HSR in patients w/ and w/o severe Covid-19 syndrome. The authors concluded that having severe Covid-19 syndrome but not diabetes increases serum HSP levels and reduces HSR. The data from the trial were well presented with proper statistics, however, there are major concerns over the conclusion and writing styles of the manuscript shown below.

1. In the conclusion, besides claiming that “critically ill patients, when compared to non-infected, presented higher eHSP and a blunted HSR”, the authors also concluded that “uncontrolled inflammation may explain the increased risk of diabetes in critically ill Covid-19 patients”. While the prior statement was backed up by trial data, the latter does not have any data support. Although the authors provided some rational discussion on why they speculate the claim to be true, without proper experimental data, it is still not appropriate to include in the conclusion.

2. The introduction and discussion part of the manuscript need to be improved. The introduction is supposed to provide sufficient background information, to present the readers with the existing knowledge of the field and to introduce the questions to be explore in this study. However, the authors currently have much of such information included in the discussion. For example, the 2nd paragraph in discussion, the authors explained in detail why eHSP is a good marker of inflammation. Such information would be helpful if stated in the introduction.

Minor concerns:

1. The authors need to explain Fig. 3 and Fig. 4 better. How does eHSP and iHSP tell different stories? It may be worth to include some information regarding this in the introduction or result part.

2.  In the discussion session, the authors spent huge effort speculating why they believe Covid-19 increases the risk of T2 diabetes. While this is an interesting hypothesis, however, this part is too long for just a part of the discussion. Please either shorten it, or include it in the main text with more data support.

Author Response

Comments and Suggestions for Authors

The manuscript by Mariana Kras Borges Russo et al. aims to compare HSP levels and HSR in patients w/ and w/o severe Covid-19 syndrome; and to understand whether diabetes has an effect on HSP levels or HSR in patients w/ and w/o severe Covid-19 syndrome. The authors concluded that having severe Covid-19 syndrome but not diabetes increases serum HSP levels and reduces HSR. The data from the trial were well presented with proper statistics, however, there are major concerns over the conclusion and writing styles of the manuscript shown below.

Author Response: We thank the reviewer for the positive observation in regards to our study and the generated results. We have addressed every point raised, and we hope you find our response satisfactory. All modifications are highlighted in yellow.

  1. In the conclusion, besides claiming that “critically ill patients, when compared to non-infected, presented higher eHSP and a blunted HSR”, the authors also concluded that “uncontrolled inflammation may explain the increased risk of diabetes in critically ill Covid-19 patients”. While the prior statement was backed up by trial data, the latter does not have any data support. Although the authors provided some rational discussion on why they speculate the claim to be true, without proper experimental data, it is still not appropriate to include in the conclusion.

Author Response: We agree with the reviewer. As also suggested by the editor, we changed this point in the text and figure. We hope that our change is satisfactory.

  1. The introduction and discussion part of the manuscript need to be improved. The introduction is supposed to provide sufficient background information, to present the readers with the existing knowledge of the field and to introduce the questions to be explore in this study. However, the authors currently have much of such information included in the discussion. For example, the 2nd paragraph in discussion, the authors explained in detail why eHSP is a good marker of inflammation. Such information would be helpful if stated in the introduction.

Author Response: We thank the reviewer for the suggestion. We moved some information from discussion to introduction. We hope this change is satisfactory.

Minor concerns:

  1. The authors need to explain Fig. 3 and Fig. 4 better. How does eHSP and iHSP tell different stories? It may be worth to include some information regarding this in the introduction or result part.

Author Response: We agree with the reviewer. We discuss the different responses/function of eHSP70 and iHSP70 (introduction).

  1. In the discussion session, the authors spent huge effort speculating why they believe Covid-19 increases the risk of T2 diabetes. While this is an interesting hypothesis, however, this part is too long for just a part of the discussion. Please either shorten it, or include it in the main text with more data support.

Author Response: As discussed before, we modified the hypotheses in the text and figure. We hope that our change is satisfactory.

Reviewer 3 Report

DearAuthors,

I have read the manuscript and I send you my comments:

1) I think that the number of enrolled patients is very low, it is a retrospective study therefore more samples must be added

2) please add more clinical data in order to evaluate a role of sex and drugs in this protein expression

3) please describe the type of diabetes and the drugs used in these patients, in order to estimate the possibile activity of these drugs in the protein expression

4) please add the time set of protein expression before and after covid-19

Author Response

Comments and Suggestions for Authors

DearAuthors,

I have read the manuscript and I send you my comments:

1) I think that the number of enrolled patients is very low, it is a retrospective study therefore more samples must be added

Author Response: We understand the reviewer concern. However, despite the sample size, including more patients is not feasible, as, fortunately, ICU admissions due covid-19 are very low at this time. However, this is a prospective cohort study, and all patients were prospectively included.

2) please add more clinical data in order to evaluate a role of sex and drugs in this protein expression

Author Response: As suggested by the editor, we added additional limitations of our work. As follows:

"Finally, low sample size, inability to stratify data by sex and inability to account for previous medications were also a limitation in this work."

3) please describe the type of diabetes and the drugs  used in these patients, in order to estimate the possibile activity of these drugs in the protein expression

Author Response: Only subjects with type 2 diabetes mellitus were included. Regarding the diabetic drugs: unfortunately, this information is not precise because a high number of subjects were diagnosed at the time of hospital admission, based on their HbA1c, thus not taking drugs. For this reason, discussing the possible effect of drugs, in our opinion, would be inadequate. We understand the reviewer concern, but this is a limitation of our work. Again, we added this as a limitation, in the text. In addition, we added a new table with all information regarding drugs, as supplemental material.  

4) please add the time set of protein expression before and after covid-19

Author Response: Unfortunately, there is no data from patients, in regards to protein expression, before covid-19. They were not regular patients from the hospital. Unfortunately, we have no data previous to the disease. 

Round 2

Reviewer 2 Report

The authors have addressed all of my comments, I am good with this manuscript being published.

Author Response

We appreciate the time and effort spent by the referee in considering our paper and generating comments, suggesting revisions and corrections. We would like to take this opportunity to thank for the valuable contribution made to improvement of our manuscript.

Reviewer 3 Report

Dear Authors,

I have read tthe manuscript and the responses. I think that more data are necessary to consider this manuscript as acceptable.

Author Response

We appreciate the time and effort spent by the referee in considering our paper and generating comments, suggesting revisions and corrections. We would like to take this opportunity to thank for the valuable contribution made to improvement of our manuscript.

We also added the limitations of the work to the conclusion.